# Catching Cancer Early: The Importance of Dermato-Oncology Screening

**DOI:** 10.3390/cancers15123066

**Published:** 2023-06-06

**Authors:** Laura Stătescu, Elena Cojocaru, Laura Mihaela Trandafir, Elena Ţarcă, Mihaela Camelia Tîrnovanu, Rodica Elena Heredea, Cătălina Iulia Săveanu, Bogdan Marian Tarcău, Ioana Adriana Popescu, Doru Botezat

**Affiliations:** 1Department of Dermatology, “Grigore T. Popa” University of Medicine and Pharmacy, 700115 Iasi, Romania; laura.statescu@umfiasi.ro; 2‘Saint Spiridon’ County Emergency Clinical Hospital, 700111 Iasi, Romania; 3Department of Morphofunctional Sciences I–Pathology, “Grigore T. Popa” University of Medicine and Pharmacy, 700115 Iasi, Romania; elena2.cojocaru@umfiasi.ro; 4Department of Mother and Child Medicine–Pediatrics, “Grigore T. Popa” University of Medicine and Pharmacy, 700115 Iasi, Romania; laura.trandafir@umfiasi.ro; 5Department of Surgery II-Pediatric Surgery, “Grigore T. Popa” University of Medicine and Pharmacy, 700115 Iasi, Romania; 6Department of Mother and Child Medicine–Obstetrics, “Grigore T. Popa” University of Medicine and Pharmacy, 700115 Iasi, Romania; mihaela.tirnovanu@umfiasi.ro; 7Department of Clinical Practical Skills, “Victor Babeş” University of Medicine and Pharmacy, 300041 Timişoara, Romania; 8Surgical Department, Discipline of Preventive Dentistry, Faculty of Dental Medicine, “Grigore T. Popa” University of Medicine and Pharmacy, 700115 Iasi, Romania; 9Department of Preventive Medicine and Interdisciplinarity—Behavioral Sciences, Faculty of Medicine, “Grigore. T. Popa” University of Medicine and Pharmacy, 700115 Iași, Romania

**Keywords:** skin cancer, monitoring, screening, malignant melanoma, squamous cell carcinoma, basal cell carcinoma

## Abstract

**Simple Summary:**

Skin cancers, either melanocytic (malignant melanoma) or nonmelanocytic (squamous cell carcinoma, basal cell carcinoma), unlike most types of cancers which are not accompanied by visible signs or symptoms in early stages, could be easily identified through screening. Patients at high risk would benefit the most from skin cancer screening programs. There is no consensus on the long-term monitoring of patients with skin cancers. Based on the clinical experience of the physician and the literature data, the advantages and disadvantages of different monitoring intervals must be assessed to provide the patient and the health system with an efficient screening.

**Abstract:**

The European Society for Medical Oncology experts have identified the main components of the long-term management of oncological patients. These include early diagnosis through population screening and periodic control of already diagnosed patients to identify relapses, recurrences, and other associated neoplasms. There are no generally accepted international guidelines for the long-term monitoring of patients with skin neoplasms (nonmelanoma skin cancer, malignant melanoma, precancerous—high-risk skin lesions). Still, depending on the experience of the attending physician and based on the data from the literature, one can establish monitoring intervals to supervise these high-risk population groups, educate the patient and monitor the general population.

## 1. Introduction

Early diagnosis and long-term monitoring of oncological patients is one of the constant concerns in medical oncology, being part of the field of cancer survivorship, according to the European Society for Medical Oncology Expert Consensus [1].

Dermato-oncology represents a special chapter in dermatological and oncological pathology, both through the implications related to interdisciplinarity and through the major psychological impact on the patients included in this category. Given the fact that dermato-oncological pathology is vast, we will refer to the most common situations in dermatological clinical practice: malignant skin tumors (malignant melanoma (MM)), basal cell carcinoma (BCC), squamous cell carcinoma (SCC), pre-carcinomatous skin lesions, or lesions with a high risk of malignant tumor association (lentigo, dysplastic nevi, actinic keratoses (AKs)).

Population-based cancer screening is a method used in the early detection of neoplasms, aiming to reduce mortality from oncological diseases in the general population. This method also refers to the long-term monitoring of patients already diagnosed with a neoplasm for an early diagnosis of relapses, recurrences, or other associated neoplasms, which may appear during the patient’s life.

Oncological screening in dermatology is required in the following scenarios:Early diagnosis and monitoring of patients with cutaneous neoplasmsHigh-risk population (Table 1)Cutaneous paraneoplastic disorders, i.e., Leser-Trélat syndrome, complex genetic syndromes, pyoderma gangrenosum, lymphocytic infiltrates

## 2. Malignant Skin Tumors

### 2.1. Malignant Melanoma

The annual incidence of MM in Europe varies from 3–5/100.000 inhabitants in the Mediterranean countries to 12–35/100.000 inhabitants in the Nordic countries [3]. The incidence in Australia and New Zealand exceeds 50/100.000 inhabitants [3].

The mortality and incidence rates of MM are higher in the countries of Eastern Europe compared to Western Europe, the difference being efficient prevention and a much earlier diagnosis in Western Europe [4].

The use of dermatoscopy, videodermatoscopy, and the “ugly duckling” sign for self-examination and, respectively, population screening, allow the early identification of as many cases of MM as possible [5].

Friedman et al. originally described the ABCD criteria to facilitate the clinical diagnosis of MM: asymmetry, border irregularity, color variegation, and diameter of at least 6 mm [6]. To improve the diagnostic sensitivity, ABCD was expanded to the ABCDE criteria, where E stands for evolutionary change [7].

The ABCDE criteria, however, fail to recognize that MMs under 6 mm in diameter exist and their incidence has increased [8], and that in situ or very early MMs may lack all these criteria. The limitations of the ABCDE criteria and clinical examination can be improved by dermoscopy, regardless of the diagnosis algorithm used [9].

Dermoscopy increased the sensitivity of the naked eye examination by 18% (95% CI 9–27%; *p* = 0.002), reaching 90% sensitivity (95% CI 80–95%) [10].

Protocols for long-term monitoring of patients with MM differ worldwide, so there is no globally accepted guideline. The primary objective of long-term monitoring is the identification of possible curable locoregional recurrences, as well as the identification of potentially associated primary MMs.

For patients already diagnosed with MM, follow-up is lifelong, as in evolution, there is a significant risk for primary MM (3.5–4.5% of cases) or local recurrences (approximately 4%—higher in the first 2–5 years after the initial diagnosis, or in the case of thick or ulcerated lesions or lesions located on the head, trunk neck or calves) [11]. Monitoring also allows the diagnosis of ultra-late recurrences (more than 15 years after the primary diagnosis), a possible and unpredictable risk [12]. Also, screening among these patients allows the diagnosis of other cutaneous or extracutaneous malignant tumors, the risk among them being increased compared to the general population [2,6].

Locoregional metastases can be identified in the case of patients with stage I or II MM, and distant metastases in the case of those with stage III or IV MM [3]; these patients are redirected to the oncologist to establish the therapeutic strategy.

Artificial intelligence has similar or even better performance (higher sensitivity and specificity) than dermatologists in diagnosing MM based on dermoscopic images [13]. Such computer algorithms can aid the physician in making the correct diagnosis and are also integrated into videodermoscopy devices.

Total body photography (2D/3D) combined with sequential digital dermatoscopic imaging (SDDI) is a very efficient tool for melanoma screening, as it facilitates monitoring all melanocytic lesions of a patient from both macroscopic and dermoscopic perspectives [14]. 

Patients at high risk would benefit the most from such technology, which allows the dermatologist to accurately and quickly assess the stability of all nevi and improve the benign-to-malignant ratio of excised melanocytic lesions. Despite these advantages, total body photography and SDDI fail to detect hypopigmented melanomas and do not allow monitoring of melanocytic lesions on the scalp, palms, soles, body folds, and genitalia. 

Routine imaging is recommended only for stages IIB to IV, 1–4 times/year, for 3–5 years, and can include chest radiograph, whole-body PET-CT, chest-abdomen-pelvis CT, or brain MRI [15]. Sentinel node-positive patients who didn’t undergo complete lymph node dissection should regularly be examined by ultrasound every four months in the first two years and then twice annually for another three years [15]. 

Currently, there is no consensus on using blood tests as a screening method in patients with MM who underwent complete surgical excision. Serum S100B protein is a moderately sensitive—61.35% (95% Confidence Interval (CI) (48.90, 73.80) and highly specific—87.30% (95% CI (81.10, 93.49)) melanoma recurrence biomarker, but because increased levels of S100B can result from several other conditions (cardiovascular disease, chronic kidney disease, liver cirrhosis, vitiligo, etc.), it would prove useful mainly in cases with a high risk of relapse [16]. In comparison to S100B, lactate dehydrogenase (LDH) has a lower sensitivity—33.93% (95% CI (17.21, 50.65)), but a similar high specificity—90.70% (95% CI (84.89, 96.51)) for cutaneous melanoma [16]. Both elevated markers are negative predictors in patients with metastatic melanoma [16]. Neither S100B nor LDH is a reliable predictor of the sentinel node status [17]. 

Melanoma-pancreatic carcinoma syndrome, also known as familial atypical multiple mole melanoma syndrome (FAMMM), is a rare inherited syndrome caused by CDKN2A mutations, which have a prevalence of <0.1% in the whole population [18]. Given this rare association of neoplasms, patients with pancreatic carcinomas or CDKN2A mutations could benefit from a total body skin examination. 

Vice-versa, melanoma patients could benefit from pancreatic carcinoma screening. The Dutch Working Group on Melanoma also considers MM screening in patients with first- and second-degree relatives with melanoma-pancreatic carcinoma syndrome or CDKN2A mutation, 1–2 times per year, starting at 12 and 20 years old, respectively [19].

Patients with MM should be made aware that their first-grade relatives are at an increased risk for MM and require a total body skin examination.

Patients and their caregivers should also be regularly counseled on how to perform life-long, between in-office visits and monthly self-examination of the entire skin, mucous membranes, and lymph nodes.

In an Italian study, patients who performed regular skin self-examinations (SSE) developed thinner melanomas than those who did not (0.77 mm vs. 0.95 mm) [20]. 

SSE reduced the risk of melanoma incidence in a study on 1199 Caucasian residents [21]. The results of this study suggested a lower risk of advanced disease in melanoma patients. However, the follow-up period was only five years [21]. The vast majority of melanoma survivors performed SSE at least monthly, but only a few (13.7%) performed a thorough SSE, and only about a quarter did it more than once a month [22].

Educational programs for school-age children provide the basic knowledge needed to limit the risk of developing MM, and children grow into educated patients who would better know when a skin examination is necessary [23,24]. Such educational interventions have proven more effective in younger school children than in adolescents [25].

In the case of patients who do not have easy access to a dermatologist or if the waiting times for a visit are too long, they can acquire a mobile and accessible dermatoscope, which can be attached to the smartphone’s camera. The resulting dermoscopic image, together with the macroscopic clinical one, can thus be uploaded online for a teledermatology service or even be analyzed by artificial intelligence. However, this option is not as reliable as a live consultation and should be employed only in-between physician office visits.

To overcome the shortage of dermatologists and the long waiting lists in some countries, general practitioners have been successfully involved in the screening of MM after following educational programs on dermoscopy and clinical diagnosis [26].

Populational melanoma screening could also be guided by the results of a questionnaire aiming to identify patients at intermediate/high risk, proving better efficacy than occasional public screening campaigns [27]. Despite initially promising results, which reported a decline of 48% in melanoma mortality over five years, long-term follow-up of a general population screening intervention in Germany eventually reported similar mortality rates to baseline [28]. 

The screening interval is controversial as it differs from one country to another, but also from one specialty to another, being greatly influenced by the practical experience of each physician.

Clinical studies have indicated that 82% of MM recurrences appear in the first five years from the primary diagnosis, with a maximum in the first two years [29], so that in the first two years, 1–4 visits/year is proposed (depending on the presence of severity factors: location, tumor thickness, personal history, etc.). After the first two years, monitoring is for life, at 6 or 12 months or upon request, depending on the number of melanocytic lesions (typical/atypical) and the number of primary MM and their characteristics.

The visit to the dermatologist involves an update on the patient’s medical history, complete clinical examination, symptoms (fatigue, headache, cough, weight loss, etc.), the appearance of new skin tumors, and changes in pre-existing skin lesions. Examining cicatricial lesions and the peri-cicatricial area is important to identify recurrences or in-transit metastases. An examination of the lymph node groups is conducted, and paraclinical and imaging investigations can be recommended. At each visit, the patient should be reminded of the rules regarding sun protection and its role.

There are numerous controversies regarding the role of screening in managing patients with MM. Advantages versus disadvantages are discussed in Table 2, both in terms of the health system and the patient.

MM overdiagnosis refers to the diagnosis of MM that would have never caused symptoms or any harm if it had remained undetected and untreated. Given the technological and theoretical advances in skin imaging techniques, which increase the sensitivity and specificity of MM diagnosis and the more aggressive screening, MM overdiagnosis has become a complex problem [30]. Patients may face unnecessary treatments, which can pose risks and cause physical and psychological harm. However, accurately distinguishing between low-risk and high-risk MM is difficult. To combat the overdiagnosis, screening only high-risk populational groups and molecular and genetic tumor characterization could prove useful.

**Table 2 cancers-15-03066-t002:** Possible advantages and disadvantages of melanoma screening and their implications.

Melanoma Screening			
Possible Advantages	Implications	Possible Disadvantages	Implications
Early diagnosis of primary skin tumors	Lower morbidity, mortality and associated costs	Overdiagnosis; Negative psychological impact on the patient	Anxiety and possibly depression associated with repeated screening visits
Identification of in-transit metastases	Evaluation of the surgical approach—better prognosis The volume of the metastases at the time of diagnosis has prognostic significance [31]	Negative impact on the health system	Overuse of human and material resources
Identification of adenopathies	The number and maximum diameter of lymph nodes influence the cure rate in stage II [31]		
Identification of distant metastases	Evaluation of the surgical approach—better prognosis		

Regardless of the discussions related to the negative impact, MM screening has proven useful both in the early diagnosis of the disease and in the long-term management of patients already diagnosed with MM.

### 2.2. Basal Cell Carcinoma

BCCs are the most common neoplasms, not only of the skin but of the entire human body, occupying the first place among nonmelanoma skin cancers (NMSCs) (70%), followed by squamous cell carcinomas (25%) [32]. Despite being about three times more common than SCC, BCC does not cause high mortality because it rarely metastasizes with an incidence of 0.0028–0.55% [33].

Worldwide there is no exact data on the real incidence of BCCs, but it is considered much undervalued. As per the existing statistics, there is an annual increase of approximately 5.5% in the incidence of BCC globally [34], and in Europe, 30–50% of cases are unreported in the registers [35]. In the United States of America, the incidence of BCC is about 525/100.000 person-years [36]. The incidence of BCC varies in the rest of the world: Australia > 1000/100.000 person-years; Africa < 1/100.000 person-years; England 76.21/100.000 person-years [34]. Most reported cases of BCCs are diagnosed in patients between 40 and 79 years old [37], and the incidence is higher in men [36].

40–50% of patients with primary BCC will develop at least one more lesion in the next 5 years, which requires careful monitoring [38]. The cumulative risk 5 years after the first diagnosis is about 29.2%, with a maximum risk in the first 6 months after the first diagnosis of BCC [39]. The risk of occurrence for the second BCC is 30% higher among men than women [39].

There are several BCC risk factors recognized, such as skin phenotype (Fitzpatrick I and II), phenotypic, genotypic, and environmental factors. However, there is still no clear relationship between ultraviolet (UV) exposure and various types of BCC.

One of the major risk factors, which can be reduced through proper education of the population, is skin photoaging. Patients with actinic keratoses have a higher relative risk of developing BCCs (Hazard Ratio 4.4, 95% CI (4.1; 5.0)) [40]. Skin elastosis lesions and solar lentigo lesions are important risk factors. UV exposure is known to be the major risk factor for NMSC. Still, its role is different: for SCC, the cumulative dose and childhood sunburns are important, while for BCC, intermittent intense exposure and sunburns at any age matter most [41,42].

Other major risk factors are iatrogenic immunosuppression, Human Immunodeficiency Virus (HIV) infection, cutaneous non-Hodgkin lymphomas, psoralen plus ultraviolet A radiation (PUVA) therapy, photosensitizing therapies, Ultraviolet B (UVB) phototherapy or occupational factors.

The diagnosis of BCC is clinical, supported by dermatoscopy and possibly skin ultrasound or confocal microscopy (especially in research centers). High-resolution optical coherence tomography can also be performed.

Case management is dictated by the clinical image, the patient’s status, the approved treatment possibilities, and the practitioner’s experience.

General population screening is recommended annually, especially among patients with major risk factors present (over 10 AKs, solar elastosis, solar lentigo) or on request when high-risk lesions develop. Screening is especially aimed at patients over 60 years of age, given that 80% of cases are diagnosed in this age group, but also at patients aged 30–49, the group in which the greatest increase in incidence has been observed. 

For patients previously diagnosed with a BCC, the proposed post-therapeutic screening interval is 3 months in the first year, 6 or 12 months in the next four years, and then annually or on request.

The role of the dermatologist is to identify the risk factors, to perform thorough clinical examination (AK, solar elastosis, solar lentigo) and dermatoscopy of suspicious lesions, and to educate the patients, with an emphasis on the prophylaxis measures.

### 2.3. Squamous Cell Carcinoma 

SCC has a metastasis rate of 5% and a recurrence rate of 8% after five years from diagnosis [43,44].

Worldwide data indicate an increasing number of cases: England 23.73/100.000 person-years, Switzerland 28.9/100.000 person-years, USA 290/100,000 person-years, Australia 387/100.000 person-years [34]. The risk among immunosuppressed patients, including acquired immunodeficiency syndrome—AIDS and PUVA therapy, is higher, with reports of 65–250 times increased risk of SCC in solid organ transplant recipients [45]. Solid organ transplant recipients who develop SCC also commonly express HPV types 8, 9, and 15 [46], but HPV could be involved only in the induction and not in the maintenance of SCC [47]. Patients with defects in the cell-mediated and humoral immunity, such as in chronic lymphocytic leukemia, have an 8–10 fold increased risk for developing SCC [48,49]. Other important risk factors are age over 70, male gender, and skin photoaging [50]. 

The positive clinical diagnosis is based on the clinical appearance of the lesion, and its history, corroborated with dermatoscopic images. High- or ultra-high frequency ultrasound can aid in the diagnosis, as it is a fast, accessible, and safe dermatological imaging technique that provides data on tumor size, depth, and vascularization (Doppler mode). Due to the lack of specific diagnostic criteria and associated SCC inflammation and hyperkeratosis, ultrasound can only be used as a complementary diagnostic technique, and its main use is to characterize the local aggressiveness of the tumor [51]. 

In vivo confocal microscopy or high-definition optical coherence tomography can also be performed, especially in specialized centers or clinical trials, because of its increased costs and difficulty of interpretation. Radiography, magnetic resonance imaging, and computer tomography have an important role in identifying the presence of metastases.

As with other skin neoplasms (BCC, MM), there is no worldwide consensus regarding the role and frequency of screening for patients with SCC. These intervals are also determined according to the clinical image, the status of the patient, the possibilities of monitoring, and the experience of the practitioner.

For SCC screening, the same intervals as for BCC are proposed. Annually for the general population, especially those at major risk. For patients already diagnosed, 3 months in the first year post-diagnosis, 6 months or 12 months in the next four years, then annually or on demand, life-long.

### 2.4. Pre-Carcinomatous/High-Risk Skin Lesions

**Lentigines** are hyperpigmented lesions that do not show changes in color intensity in the absence of UV exposure (differential diagnosis with ephelides), produced by the proliferation of melanocytes secondary to exposure to UV (natural/artificial), with pigment concentration in the epidermal basal layer [52]. They are represented by lentigo simplex, solar lentigo, post-phototherapy lentigo, lentiginosis profusa, segmental lentigines, or familial lentiginosis syndromes. 

Familial syndromes have an autosomal dominant inheritance (Peutz-Jeghers, PTEN, LEOPARD, NAME, etc.), most of them being caused by mutations of RAS-MAP (the rat sarcoma-mitogen-activated protein) kinase and mTOR (the mammalian target of rapamycin) signaling pathway [53] and being associated with increased risk of developing lentigo maligna melanoma [54]. 

Lentigines can also be located on the mucous membranes (labial or genital), being sometimes associated with genital lichen sclerosis [53].

Most lentigines are benign, but some can be associated with neoplasms. Post-photochemotherapy lentigines are similar to ephelides or lentigo simplex from a pathophysiological point of view. Still, melanocytes may present nuclear atypia, more numerous dendrites, and more intense melanogenesis processes [53]. Therefore, post-photochemotherapy lentigines are considered potential marker lesions for developing NMSC, so they must be monitored.

**Atypical nevi** are nevi larger than 5 mm in size, with a macular component, imprecisely demarcated edges or irregular outline, asymmetrical appearance, or variable pigmentation [53].

Atypical nevi were originally described by Clark et al. in 1978, in families at high risk for MM, as distinct nevi with different clinical and histopathological features in the same individual [55]. Initially named B-K syndrome, they were also included in other syndromes such as FAMMM, dysplastic naevus syndrome(DNS) or atypical naevus syndrome(ANS), etc. [53]. Similar to common nevi, they are considered genetically determined [56].

Atypical nevi are frequently encountered under the age of 30–40 and are associated with the risk of developing multiple MMs [57]. They are frequently encountered in patients with MM (34–59% of cases) [58]; however, their rate of transformation to MM is low [57].

Histopathological studies show that approximately ¼ of MMs develop on pre-existing lesions [59] and the rest on normal skin. There are no markers or criteria for identifying lesions at high risk of transformation.

Study data indicate a relatively higher risk of developing MM in patients with atypical nevi as follows: 1 lesion—1.45×; 5 lesions—6.36× [60]; more than 10 lesions—12× [61], and among families with multiple members with atypical nevi, the risk is 85 times higher [62].

Screening for patients with lentigines/atypical nevi is proposed every 6–12 months or on request through dermoscopy, videodermoscopy, or total body photography and sequential digital dermoscopic imaging. Atypical nevi are considered major risk factors, but to a lesser extent, precursors to MM [57]. In both situations, the dermatologist should insist on prophylaxis measures.

**Actinic keratoses** are keratotic lesions characteristic of adults located in UV-exposed areas, with a low risk of transformation into SCC [63]. Histopathologically, they are characterized by epidermal dysplasia (restricted to the basal layer or affecting the entire epidermis), with hyperkeratosis, parakeratosis, hypogranulosis, and multiple nests of atypical cells at the level of the basal membrane area, but without invasion [53].

The internal risk factors that must be considered are the skin phototype (Fitzpatrick I–III) and old age. Among the external risk factors, we mention chronic exposure to UV (sun/phototherapy, etc.), leading to DNA degradation and the proliferation of genetically altered keratinocytes and local immunosuppression [53].

Based on their thickness, Olsen et al. classified AKs into 3 grades, from slightly palpable to very thick and hyperkeratotic [64].

Monitoring of patients with AKs involves global risk assessment (onset/growth rate/immunosuppressive therapies, etc.) and annual visits with the clinical and dermatoscopic evaluation of suspicious lesions, especially among patients with major risk factors.

In a study of 624 patients with at least 5 AKs on the head, treated with different local therapies, the total 4-year risk of developing SCC was 3.7% (95% CI (2.4, 5.7)), the lowest risk being associated with fluorouracil (2.2%) and the highest risk with imiquimod (5.8%) [65]. Olsen grade III AKs had SCC developing risk of 20.9% (95% CI (10.8, 38.1)), prompting close follow-up after treatment [65].

### 2.5. Cutaneous Neoplasms in Immunocompromised Patients

The immune system plays a critical role in the development of skin neoplasms, their regression, and their destruction. Monitoring and screening for skin neoplasms in immunocompromised patients is a challenge, given that:Their frequency, diversity, atypia, and aggressiveness are much higher compared to the general population;Mortality and morbidity are significant;There are no guidelines for the management of these scenarios.

Some significant risk factors must be considered to differentiate populational groups at risk:Duration of immunosuppression/age of transplantation—in transplanted patients, a 12-fold higher relative risk was observed in those over 55 years old compared to those under 34 years old [66];Fitzpatrick I–III skin phototype, pre-immunosuppression sunburn history, and the presence of AKs or solar lentigines [67,68,69].

Immunosuppression can be divided into:Primary immunosuppression: epidermodysplasia verruciformis—X-linked genodermatosis associated with HPV infection; Wiskott-Aldrich syndrome—immunodeficiency associated with autoimmunity and eczema; Netherton syndrome; neurofibromatosis type I; ichthyosiform erythroderma associated with atopy, asthma, and food allergies [53,70];Secondary immunosuppression: HIV infection; drug-induced immunosuppression (transplantation/autoimmune diseases/inflammatory diseases); patients in end-stage organ disease [53].

Epidemiological studies show a greatly increased risk of skin neoplasms among immunocompromised patients:SCC: 65–250× post-transplantation [39]; 1.5–8× other immunocompromised patient groups [71,72,73];BCC: 10× post-transplantation [74]; high-risk BCC more frequent in HIV [75];MM: 20% more frequent recurrences in patients with pre-transplant MM [76]; second most common donor-transmitted cancer [77], which is associated with severe prognosis because 80% rapidly metastasize [78,79]; 2.6× more frequent de novo MM in HIV patients [80]/chronic lymphocytic leukemia patients: 2–7× more frequent [81].

As in the general population, there are no guidelines for monitoring immunosuppressed patients. Screening is carried out depending on the clinical experience of the practitioner and the risk factors, at 6 or 12 months or on request, by clinical examination, dermatoscopy, videodermoscopy, total body photography, and sequential digital dermatoscopic imaging.

Transplant patients represent a special group that requires:Pre-transplant screening for pre-carcinomatous skin lesions and possible identification of Human Herpes Virus-8 infection [82].Post-transplant screening:
○Education (photoprotection/total body self-examination/early diagnosis)—all patients;○Dermatological screening: at least every 12 months, stratified by risk groups for all patients depending on the age of the transplantation, skin phototype, personal history of skin neoplasia, and childhood sunburns.Patients with premalignant/malignant skin lesions with low risk require annual screening, and those with high risk—screening at 3, 6, or 12 months, depending on the type of neoplasm.

## 3. Conclusions

The lack of consensus on the screening protocols for skin cancer implies that the physician has the freedom and responsibility to adjust the screening methods based on personal experience and the specialty literature.

High-risk patient groups should be well established, and effort should be put into addressing their medical needs, not only before the diagnosis of skin cancer but also during cancer survivorship. An optimal screening would positively impact the early detection of skin cancers, further reducing morbidity, mortality, and associated costs.

## 4. Future Directions

A nationwide well-established and accepted skin cancer screening protocol would be needed to systematically monitor the outcomes over time. The synergistic use of 3D total body photography, sequential digital dermatoscopic imaging, and artificial intelligence should become more widely available to facilitate the screening process. Identifying melanoma biomarkers, susceptibility genes, or combinations of both, ideally with high specificity and sensitivity, would provide a valuable risk criterion and screening addition.

## Figures and Tables

**Table 1 cancers-15-03066-t001:** Criteria for skin cancer in a high-risk population.

High-Risk Criteria	Details
Skin phototype	Fitzpatrick I-III [2]
Light eyes, light or red hair	
Tanning	Blistering sunburns/indoor tanning
Nevus count	≥50 common nevi
Pre-carcinomatous or high-risk skin lesions	Ephelides, chronic actinic skin damage, atypical nevi, giant congenital nevi
Personal history of:	
BCC	
SCC	
Pancreatic carcinoma	
CDK2NA mutation	
Immunodeficiency	Primary or secondary immunodeficiency (e.g., organ transplant recipients)
Family history of MM	First-degree relatives

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
