# Peer review of "Catching Cancer Early: The Importance of Dermato-Oncology Screening"

_cancers, 2023, doi:10.3390/cancers15123066_

Round 1

Reviewer 1 Report

Dear authors, the paper is a good contribution. You have to perform some changes especially in melanoma section.

- I suggest to add in Table 1 "organ transplant recipients" among acquired immunosuppression.

- add "i.e." after paraneoplastic disorders

- I am not sure that here "The mortality and incidence rates of MM are higher in the countries of Eastern Europe compared to Western Europe, the difference being justified by efficient prevention and a much earlier diagnosis in Eastern Europe" you correctly reported the results of the study. Please correct the sentence.

- in the malignant melanoma paragraph you should mention the role that "ABCD" had in the past (Friedman R, 1985) and how dermoscopy changed the cut off of suspicious melanocytic lesions (Argenziano G, Soyer HP. Dermoscopy of pigmented skin lesions--a valuable tool for early diagnosis of melanoma. Lancet Oncol. 2001 Jul;2(7):443-9. doi: 10.1016/s1470-2045(00)00422-8. PMID: 11905739.) untile the recent crescent reports of small diameter melanomas (Nazzaro G, Passoni E, Pozzessere F, Maronese CA, Marzano AV. Dermoscopy Use Leads to Earlier Cutaneous Melanoma Diagnosis in Terms of Invasiveness and Size? A Single-Center, Retrospective Experience. J Clin Med. 2022 Aug 21;11(16):4912.)

- the paragraphs starting "routine imaging" and "Currently, there is no consensus on using blood tests" are off topic; I think they must be reduced or deleted.

- the topic about "melanoma overdiagnosis" should also discussed (https://www.aad.org/dw/dw-insights-and-inquiries/archive/2021/epidemic-of-melanoma-or-epidemic-of-scrutiny   -  Bell KJL, Nijsten T. Melanoma overdiagnosis: why it matters and what can be done about it. Br J Dermatol. 2022 Oct;187(4):459-460

Author Response

Dear Reviewer,

Thank you very much for evaluating our manuscript. Your recommendations and comments have helped us improve our manuscript. Here we provide the requested corrections and address the comments. The changes we have made in the manuscript are highlighted in red.

  1. Dear authors, the paper is a good contribution. You have to perform some changes especially in melanoma section.

- I suggest to add in Table 1 "organ transplant recipients" among acquired immunosuppression.

Response: We added the suggested phrase in table 1.

  1. - add "i.e." after paraneoplastic disorders

Response: We corrected.

  1.  - I am not sure that here "The mortality and incidence rates of MM are higher in the countries of Eastern Europe compared to Western Europe, the difference being justified by efficient prevention and a much earlier diagnosis in Eastern Europe" you correctly reported the results of the study. Please correct the sentence.

Response: We corrected, thank you.

  1.  - in the malignant melanoma paragraph you should mention the role that "ABCD" had in the past (Friedman R, 1985) and how dermoscopy changed the cut off of suspicious melanocytic lesions (Argenziano G, Soyer HP. Dermoscopy of pigmented skin lesions--a valuable tool for early diagnosis of melanoma. Lancet Oncol. 2001 Jul;2(7):443-9. doi: 10.1016/s1470-2045(00)00422-8. PMID: 11905739.) untile the recent crescent reports of small diameter melanomas (Nazzaro G, Passoni E, Pozzessere F, Maronese CA, Marzano AV. Dermoscopy Use Leads to Earlier Cutaneous Melanoma Diagnosis in Terms of Invasiveness and Size? A Single-Center, Retrospective Experience. J Clin Med. 2022 Aug 21;11(16):4912.)

Response: We completed the paragraph according to your suggestions.

  1. - the paragraphs starting "routine imaging" and "Currently, there is no consensus on using blood tests" are off topic; I think they must be reduced or deleted.

Response: We decided to let the information.

  1. - the topic about "melanoma overdiagnosis" should also discussed (https://www.aad.org/dw/dw-insights-and-inquiries/archive/2021/epidemic-of-melanoma-or-epidemic-of-scrutiny   -  Bell KJL, Nijsten T. Melanoma overdiagnosis: why it matters and what can be done about it. Br J Dermatol. 2022 Oct;187(4):459-460

Response: We discussed the topic and added the suggested references.

Thank you again for reviewing our manuscript,

Reviewer 2 Report

Dear Authors, the topic is interesting and complex at the same time.

Overall, a good job. 

Just few considerations:

I would create subparagraphs to make the reading more fluid. Moreover, concerning the melanoma part, "In the case of patients who do not have easy access to a dermatologist or if the waiting times for a visit are too long, they can acquire a mobile and accessible dermatoscope, which can be attached to the smartphone's camera. The resulting dermoscopic image, to- 149 gether with the macroscopic clinical one, can thus be uploaded online for a teledermatology service or even be analyzed by artificial intelligence", altough being a possibility, it is harmful. So, it would be better to clarify the somehow standardized strategies from the "necessary" ones. 

Correct also the following:

-the parenthesis, introduction paragraph, line 48

- replace one of the double parentesis with [], line 53-56

Leser-Trélat syndrome

Familial atypical multiple mole melanoma (FAMMM) instead of familiar melanoma syndrome

- oncological diseases instead of pathologies, line 58 

Author Response

Dear Reviewer,

Thank you very much for evaluating our manuscript. Your recommendations and comments have helped us improve our manuscript. Here we provide the requested corrections and address the comments. The changes we have made in the manuscript are highlighted in red.

Dear Authors, the topic is interesting and complex at the same time.

Overall, a good job.  Just few considerations:

  1. I would create subparagraphs to make the reading more fluid. Moreover, concerning the melanoma part, "In the case of patients who do not have easy access to a dermatologist or if the waiting times for a visit are too long, they can acquire a mobile and accessible dermatoscope, which can be attached to the smartphone's camera. The resulting dermoscopic image, together with the macroscopic clinical one, can thus be uploaded online for a teledermatology service or even be analyzed by artificial intelligence", altough being a possibility, it is harmful. So, it would be better to clarify the somehow standardized strategies from the "necessary" ones.

Response: Thank you for your remark, we added clarifications.

  1. Correct also the following:

-the parenthesis, introduction paragraph, line 48

Response: We corrected.

3. - replace one of the double parentesis with [], line 53-56

Response: We corrected.

  1.  - Leser-Trélat syndrome

Response: We corrected.

  1. - Familial atypical multiple mole melanoma (FAMMM) instead of familiar melanoma syndrome.

Response: We corrected.

  1. - oncological diseases instead of pathologies, line 58

Response: We corrected.

Thank you again for reviewing our manuscript,